

# Seasonal and synoptic variability of diurnal currents in an upwelling system off northern Chile near 30°S

Mónica Bello[1,2], Marcel Ramos[3, 4, 5, 6], René Garreaud[1,2], Luis Bravo[3, 4], and Martin Thiel[3, 4, 5]

[1]Departamento de Geofísica, Universidad de Chile, Santiago, Chile
[2]Center of Climate and Resilience Research, Universidad de Chile, Santiago, Chile
[3]Departamento de Biología Marina, Facultad de Ciencias del Mar, Universidad Católica del Norte, Coquimbo, Chile
[4]Millennium Nucleus for Ecology and Sustainable Management of Oceanic Islands (ESMOI), Universidad Católica del Norte, Coquimbo, Chile.
[5]Centro de Estudios Avanzado de Zona Aridas, Coquimbo, Chile
[6]Centro de Innovación Acuícola Aquapacífico, Coquimbo, Chile

*Correspondence to*: Mónica E. Bello (monica@dgf.uchile.cl) and Marcel Ramos (marcel.ramos@ucn.cl)

**Abstract.** This study documents the seasonal and synoptic variability of diurnal currents in northern-central Chile (~30°S), using current measurements from four sites collected over more than one year. The study area includes a coastal upwelling center well exposed to sea wind and a large bay system (~100 km long) located just north of the above mentioned upwelling center. This bay system consists of several smaller bays with different orientations and morphologies, which affect the internal hydrodynamics and favor local recirculation patterns. Inertial oscillations in the area have a period of ~24 h, which is the same as that of the periodic wind forcing due to the sea breeze, and thus, this coupling may cause system resonance, as has been reported in other regions. The most intense diurnal currents (with amplitudes of ~30 cm s$^{-1}$) were recorded in the surface layer in one of the areas exposed to the wind and farthest from the coastline (to ~22 km). In contrast, within the bay system, which is sheltered from the wind, diurnal currents were less intense (~10 cm s$^{-1}$). Diurnal currents had higher seasonal variability in the more exposed areas than in the protected ones and were more intense in spring and summer than in autumn and winter. This was consistent with the Lagrangian measurements of the surface currents, which showed a higher diurnal energy in summer than in winter. The diurnal wind variability was modulated by the synoptic-scale circulation, which directly affected the diurnal current response. Under upwelling-favorable winds, diurnal currents were mainly forced by daily wind variations due to the sea breeze, while a sudden decrease in wind speed generated inertial oscillations that decayed with depth, especially in the area farthest from the coast. In general, the greatest variability in the diurnal currents occurs in the most exposed area to the wind and farthest from the coast, due to resonance between diurnal wind forcing and inertial oscillations, and possibly by near-diurnal internal gravity waves.



## 1   Introduction

The coastal waters of northern-central Chile (~29–31°S) exhibit features typical of a subtropical
upwelling system. The climate is dominated by the South Pacific anticyclone that favors semi-arid
conditions and the prevalence of southerly winds throughout most of the year, which promotes the
upwelling of cold waters along the coast (Strub et al., 1998; Rahn et al., 2011) and high marine
biological productivity (Fonseca and Farías, 1987). The study area corresponds to a large bay system
that extends southward for ~100 km and whose southern half consists of several smaller bays with
different orientations and morphologies that affect the internal hydrodynamics of the system (Valle-
Levinson et al., 2000; Valle-Levinson and Moraga, 2006; Moraga-Opazo et al., 2011). The embayments
within the system are Coquimbo Bay with a nearly N-S orientation open to the west (Fig. 1a), while
Tongoy Bay has an E-W orientation, is open to the north, and has its bathymetry approximately parallel
to the coastline, with sudden changes around Punta Lengua de Vaca. These differences indicate
contrasting circulation patterns (Valle-Levinson and Moraga, 2006; Moraga-Opazo et al., 2011).

In the southern part of the Coquimbo Bay system, specifically to the north of Punta Lengua de
Vaca (Fig. 1), the surface wind is a low-level jet stream with significant diurnal variation, with a
maximum at ~18:00 (local time) (Garreaud et al., 2011). Thus, strong sea-land temperature gradients
may significantly affect bay circulation patterns, particularly those related to the sea breeze, which is
quite strong in spring and summer due to intense daytime heating and nighttime cooling (Simpson et al.
2002; Hyder et al., 2011). One of the first investigations of the circulation in Coquimbo Bay was
undertaken by Valle-Levinson et al. (2000), who recorded currents for 24 h and suggested that the
vertical structure of the circulation consists of two layers, with the behavior of the surface layer
dominated by the effects of the diurnal wind and tidal variability. Later, Valle-Levinson and Moraga
(2006) found a similar surface-layer circulation pattern between the Coquimbo and Guanaqueros Bays,
which also included a pair of counterclockwise gyres, one of which was attributed to flow separation at
Punta Tortuga and the other to that at Punta Guanaqueros. A study of the circulation near Tongoy Bay
under wind conditions favorable to upwelling revealed cyclonic recirculation flowing southward near
the coast (in the eastern sector of the bay) and northward near Punta Lengua de Vaca (Moraga-Opazo et
al., 2011). Later measurements, after an extended wind relaxation period (~1 week), showed a change in
the recirculation pattern near the mouth of Tongoy Bay due to an anticyclonic regime (Ramos, pers.
comm.). Marin and Delgado (2007) proposed that the surface coastal circulation between 23°S and 30°S



has a northerly direction with magnitudes of approximately 20 cm s$^{-1}$ and that, close to the coast (<30
km off the coast) at ~30°S, there are inertial oscillations (recorded by surface drifters in the coastal area
of Coquimbo Bay), which increase the retention time of coastal waters in the area.

Diurnal wind forcing near Coquimbo (~30°S) by the sea breeze can equalize the diurnal and

inertial frequencies (Craig, 1989). At this latitude, oceanic inertial oscillations with a period of ~24 h
could resonate with diurnal wind variations (Simpson et al., 2002). In a region between ~29°S and 31°S,
there are few published studies on the variability of diurnal band currents, and these do not consider
periodic wind forcing, inertial oscillations, or their interactions. Inertial oscillations may play a dominant
role in current fluctuations in the ocean surface layer (Garret, 2001; Hyder et al., 2011). Simpson et al.
(2002) and Hyder et al. (2011) suggest that at approximately 30°N/S, there may be resonance between
the diurnal and inertial frequencies, i.e., diurnal winds could force an anticyclonic gyre that would
contribute to the energy near the inertial band. One study of the daily cycle of coastal circulation on the
interior shelf of Concepcion (37°S, Chile) during summer showed high current variability and diurnal
intensification when the daily wind cycle increased in amplitude (Sobarzo et al., 2010). Additionally, the
study identified a two-layer vertical structure consisting of a surface layer and a bottom layer with a
phase shift of ~180°, which was explained using the simple two-layer model with diurnal wind forcing
near the coast proposed by Simpson et al. (2002), where the flow is bounded by a rectilinear coastline
and includes a zonal pressure gradient in both layers. Apart from the study by Sobarzo et al. (2010), in
Chile, there are no studies assessing periodic wind forcing, inertial oscillations, and their possible
resonance in a region near 30°S. The availability of current data from a long period (~one year) from
four sites in the study area (29-31ºS) (corresponding to different levels of wind exposure) will allow us
to assess for the first time the effects of periodic and episodic wind forcing on the variability of diurnal
currents and consider seasonal and synoptic variability.

The objective of this study is to analyze the seasonal and synoptic variability in the daily cycle of

marine currents on the continental shelf and at the mouth of two internal bays in northern-central Chile
(~29–31°S). This report is organized as follows: first, there is a description of the study area, data used,
and the processing of the meteorological and oceanographic data. Then, there is an analysis and
discussion of diurnal wind forcing, followed by a general description of the mean currents, the seasonal
wind and surface current patterns, the daily current cycle and its vertical structure, and the synoptic
variability of the wind and its impact on diurnal currents and a case study on diurnal wind-driven



currents in spring. Finally, we present a summary of the main results of this study.

**2    Methods**

**2.1 Study zone and information**

The study area (Coquimbo) is characterized by a system of bays (including the bays of Coquimbo,

La Herradura de Guayacan, Guanaqueros, and Tongoy, among other smaller inlets) that represent the
largest topographic "accident" in the meridional direction (~100 km) in northern-central Chile, which
modifies the morphology and orientation of the coastline (Fig. 1). Additionally, it is part of one of the
most intense upwelling centers north of Punta Lengua de Vaca in Chile (Strub et al., 1998; Shaffer et al.,
1999; Shaffer et al., 2004). In this study area (29−31°S, Chile), oceanic measurements are available for
2009 and 2010, which include current data (Islote Pájaros, Talcaruca, and Coquimbo and Tongoy Bays)
(Fig. 1a) and coastal surface wind time series at Islote Pájaros (IP), Coquimbo (COQ), Punta Lengua de
Vaca (PLV), and Talcaruca (TCR) (Fig. 1b). Marine current measurements were obtained using an
ADCP-RDI (Acoustic Doppler Current Profiler, model 300 kHz) moored at four different sites, two of
which were located on the continental shelf at Islote Pájaros and Talcaruca, which we will define in this
study as the exposed system. Despite considering these two locations to be in the exposed system, it
should be noted that IP is a small island located 45 km north of Coquimbo and approximately 22 km off
the continent. The other two moorings were placed at the mouth of the Coquimbo and Tongoy (TON)
Bays. ADCPs were programmed with a vertical resolution of 4 m and a sampling interval of 30 minutes
(Table 1). As additional information, Lagrangian measurements of surface currents in the Coquimbo
Bay area (~30°S) were available through buoys equipped with a satellite transmitter GLOBALSTAR
(SmartOne B model) and a GPS that recorded position and trajectory every 35 minutes for 16 days in the
summer (January, 2012) and in the winter (July, 2011).

**2.2 Data analysis**

Time records of ocean currents every 30 minutes were hourly averaged. To examine the energy

distribution over different frequencies, a complex spectral analysis was performed to currents and winds
at IP and TCR. This technique allows the vectorial time series to be decomposed into an anticyclonic
part (counterclockwise gyre) and a cyclonic part (clockwise gyre) at different frequency bands (Mooers,



1973). Additionally, time records of all the surface currents in the study area were analyzed using a
harmonic analysis (Emery and Thomson, 2001) at a frequency of 1.0 cpd (diurnal band) to obtain the
major and minor semi-axis and the inclination, which are represented as current ellipses in the diurnal
frequency (Fig. 1).

Diurnal variability in wind and currents was defined by the frequency band between 0.73 and 1.60

cpd and obtained by applying a Cosine-Lanczos bandpass filter (Sobarzo et al., 2010). This filter
includes variability in the diurnal current that may be due to diurnal wind forcing, inertial oscillations,
and/or possible tidal influences. The tidal constituent that could have an impact on this area is the solar
diurnal $K_1$ (0.997 cpd) component, which here has an amplitude of ~1.0 cm s$^{-1}$. According to previous
studies on the continental shelf (Ramos, 1999; Bravo et al. 2013), the $K_1$ component is clearly lower
than the amplitude of periodical wind-forced currents or those forced by intermittent high-intensity
events. For this reason, the analysis focused not on tidal current forcing but on the effect of diurnal wind
forcing. Finally, to differentiate diurnal fluctuations from currents due to diurnal wind forcing or inertial
oscillations, complex demodulation was applied (Pollard and Miller, 1970; Simpson et al., 2002;
Sobarzo et al., 2010).

**3   Results and discussion**

**3.1 General description of the winds and currents**

A general view of the ocean circulation in our study region can be obtained from the vertical

profiles of the mean (average over all measurement period) zonal and meridional current components
presented in Fig. 2. The two stations on the continental shelf (IP and TCR) revealed an eastern boundary
current system characterized by an equatorward flow in the surface layer and a subsurface current
toward the Pole (Strub et al., 1998). The IP station was characterized by a flow toward the northwest that
extended from the surface until ~35 m deep and a flow toward the southeast below 45 m. At TCR, the
results showed a mean northward flow in the surface layer (12–30 m) of 6.3 cm s$^{-1}$ at the surface (12 m),
and the bottom layer was represented by a mean southward flow of 3.4 cm s$^{-1}$ at approximately 60 m.
The TCR station was located close to the coast (~2 km), which apparently constrains the development of
strong zonal currents. As a result, the flow aligns with the coast and the bathymetry, and thus, the zonal
components are close to zero. The difference in the flow orientations of the surface layers of the IP and



TCR stations was due to topographic effects and differences in wind forcing. At the entrance of the
Coquimbo and Tongoy Bays, the mean circulation was similar in direction and magnitude across the
water column. In Coquimbo Bay, the mean circulation had a predominant southward orientation, and the
zonal components were close to zero from surface to 60 m. In Tongoy Bay, the mean circulation was
mainly directed toward the southwest.

To sumarize the regional circulation (Fig. 1), we include the average currents for a representative

depth in the surface layer (~10 m) and lower layer (~60 m). The circulation on the continental shelf (IP
and TCR) is characterized by an equatorward flow in the surface layer and a subsurface current by a
southward flow (Strub et al., 1998), which we describe as a system exposed to the prevailing winds. At
the entrance of the bays (COQ and TON), the flows are characterized by local recirculation patterns. In
this case, the bays are described as systems sheltered from the sea wind. The surface circulation
observed at the entrance of the bays (Fig. 1a) agrees with other studies that have proposed other
recirculation systems for both embayments. Valle-Levinson and Moraga (2006) proposed two gyres in
Coquimbo Bay, and Moraga-Opazo et al. (2011) proposed that circulation in Tongoy Bay is part of a
cyclonic gyre formed by flow separation at Punta Lengua de Vaca. However, several gyres may form in
this system of bays since Valle-Levinson and Moraga (2006) described the presence of cyclonic gyres to
the south of the Coquimbo and Guanaqueros Bays, which were attributed to northward flow separation
on the continental shelf at the Tortuga and Guanaquero points.

The differences in current intensities and directions between the two bays were due to wind

forcing, orientation, and the size of the bays (Fig. 1a). The surface currents in Tongoy Bay were greater
than those in Coquimbo Bay due to intense wind forcing at Punta Lengua de Vaca (PLV), which favors
a local atmospheric jet (Garreaud et al., 2011; Rahn et al., 2011), as described previously. Moraga-
Opazo et al. (2011) estimated that one gyre may form in Tongoy Bay. In contrast, in Coquimbo Bay,
two gyres may occupy much of the bay (Valle-Levinson and Moraga, 2006). The difference in the
number of gyres that are formed in each bay indicates that there are different factors and drivers of local
recirculation patterns.






### 3.2 Spectral variability in the winds and currents of the region

A rotational spectral analysis applied to the winds observed at Islote Pájaros (Fig. 3a) and Talcaruca (Fig. 3b) showed high energy levels in the diurnal (D) and semidiurnal (SD) bands; the diurnal band had predominantly counterclockwise rotation. Previous studies of wind forcing in the area of Coquimbo had shown that the wind is characterized by a clear daily cycle with maximum speed in the afternoon directed toward the northeast (Rahn et al., 2011). Diurnal variability in the zonal and meridional wind components was quantified using a 24 h harmonic least-squares analysis, a 30% of the total variance was observed for COQ and PLV, while 13% was found for IP and TCR (Table 2). The spectral analysis of the surface currents at IP (8 m) and TCR (12 m) showed high energy levels in the diurnal and semidiurnal bands. The diurnal frequency dominated and had predominantly counterclockwise rotation, rather than clockwise (Fig. 3c, d). This behavior was maintained in the distribution of the bottom current energy for all locations (Fig. 4) and is intense down to ~50 m. Since we are interested in studying the variability of the diurnal currents due to wind forcing, we will focus on the results of this frequency and ignore the semidiurnal band, which was reviewed by Bravo et al. (2013).

The strong increase in the spectral energy of the currents, observed in the diurnal band (Fig. 3), may be explained by diurnal wind forcing. A coherence squared analysis between the wind and surface currents (not showed) corroborated a correlation in the diurnal band, which was significant at the 95% confidence level. However, part of this energy increase may be due to the generation of inertial oscillations, as their period in the study area (30°S) is ~24 h, and they could be resonating with diurnal wind forcing (Simpson et al., 2002; Hyder et al., 2011). These oscillations are anticyclonic rotations (counterclockwise) that occur when the wind suddenly changes in magnitude or direction. Off the coast of Chile, inertial currents have been identified near 30°S by Lagrangian observations (Marin and Delgado, 2007; Chaigneau et al., 2008). In southern-central Chile, specifically on the continental shelf of Concepcion (37°S), inertial oscillations have also been inferred from Eulerian measurements (Sobarzo et al., 2007; Sobarzo et al., 2010). A rotational spectral analysis applied to the currents observed in the water column showed that IP was the location with the highest energy in the diurnal band (Fig. 4). The extent of the variance explained by the meridional and zonal components of the surface currents in the diurnal frequency band was highest at IP, and these two components each accounted for 14% of the variance (Table 3).



To obtain a general view of the daily wind behavior, an hourly average of the U and V

components was obtained by considering almost ~1 year of data (December 1, 2009 until October 22,
2010). The hodographs (using the least square method) show anticyclonic gyres with different
amplitudes and inclinations (Fig. 1c). At TCR, the hodograph showed a marked north-south orientation
parallel to the coastline, as a consequence of its topographical constraints. At IP, the ellipse was slightly
oriented toward the northeast with a greater amplitude than that observed at TCR; this was due to the
intense wind forcing in the more oceanic region (i.e., the exposed system). The ellipses at PLV and COQ
were predominantly oriented toward the northeast due to the afternoon breeze toward the continent
(Garreaud et al., 2011).

The diurnal surface currents showed an elliptical tendency and an anticyclonic gyre at all the

stations (Fig. 1a). At IP, the main axis of the ellipse deviated toward the northwest, and the orientation
of the TCR ellipse was similar to the alignment of the TCR wind hodograph (Fig. 1c), with a north-south
tendency that was aligned with the coastline. The ellipse at Coquimbo Bay was perpendicular to the
coast, and the major axis of the ellipse at Tongoy Bay was aligned with the topography and bathymetry
of PLV. Additionally, the ellipses representing the bottom layer also had elliptical shapes and
anticyclonic gyres (Fig. 1b), but with much lower amplitudes than those recorded for the surface
currents. On the other hand, at all the sites the diurnal currents had an anticyclonic polarization (Fig. 2c).
Note that in the southern hemisphere the inertial currents have an anticyclonic rotation, that is, a
polarization less than 1.

**3.3 Seasonal variations in the winds and surface currents**

To obtain an overview of the seasonal behavior of the wind (Fig. 5) and currents (Fig. 6), hourly

averages of the U and V components for each month were obtained. Note that at IP the wind was
calculated with data from 2012 and 2013, but the temporal variability in that period was similar to the
data available for 2010. As previoulsy documented (Garreaud et al., 2011; Rahn et al., 2011) the wind at
IP and PLV has a marked seasonal cycle, with higher intensities in spring and summer and lower ones in
autumn and winter. At TCR, the wind had a weaker seasonal cycle but again showed a higher intensity
in spring. At COQ, a clear seasonal wind cycle could not be identified. However, there was an increase
in the magnitude of the spring and summer periods (November 2009 to February 2010). The wind curl
that developed in this area was responsible for the wind pattern at COQ, as described previously



(Garreaud et al., 2011; Rahn et al., 2011; Bravo et al., 2016).

The seasonal cycle of the mean surface current (Fig. 6) was more complex and less marked than
that of the wind. We also found that in the exposed system, the temporal variations of the surface
currents showed signs of intraseasonal influence. In May 2010, a noticeable change in the direction of
the surface currents of the exposed system (IP and TCR) was identified (Fig. 6). This may be due to the
passage of a low frequency perturbation of equatorial origin that propagated as a coastal-trapped wave
toward the south (Shaffer et al., 1997). This interpretation is based on a review of the sea level
measurements at Coquimbo, where intraseasonal disturbances were identified between February and
April 2010 (not shown). Due to the small signal-to-noise ratio, a much larger (several years) data record
would be needed to fully characterize the seasonal cycle of the surface currents. Nonetheless, we found
that in the exposed system, surface current intensity was higher in spring and lower in autumn (TCR)
and winter (IP). The maximum current intensities agreed with the maxima of the wind during the spring.
At the entrance of the bays, a seasonal pattern was not identified for the surface currents. However, at
COQ, an increase in the current intensity during the summer months was identified in response to the
increase in the wind intensity during the same period. The temporal variation in the surface currents in
Tongoy Bay did not agree with the wind at PLV, possibly because the TON measurements were close to
the cyclonic recirculation center found at the entrance of the bay as described by Moraga-Opazo et al.
(2011), which apparently responds to the wind stress curl in the southern zone of the Coquimbo Bay
system.

Trajectories obtained by satellite-tracked surface buoys (note that measurement periods were not
the same as those of the ADCPs) that were placed in front of the Coquimbo Bay system showed marked
differences in the area associated with inertial oscillations between summer (Fig. 7a) and winter (Fig.
7b), with stronger influence during summer. In the study area, the maximum winds are in spring and
summer, which suggests that the most intense inertial oscillations should occur during this time of the
year. Additionally, Rahn and Garreaud (2013) reported bi-monthly mean wind speed fields that showed
that wind intensity is higher during austral summer than in winter at ~30°S in a band extending from the
coast to 76°W. This result agrees with the study by Chaigneau et al. (2008), in which they found inertial
currents in the southern hemisphere of approximately 8 to 10 cm s$^{-1}$, with higher amplitudes during
summer rather than winter. In the study area, Marin and Delgado (2007) measured currents near the
coast (<30 km off the coast) using surface drifters and identified a northward flow with inertial effects





during January 2005 and 2006. Most of the records over 50 cm s$^{-1}$ were associated with current drift
toward the NW during the presence of coastal jets and filaments.

**3.4 Diurnal variations in the winds and surface currents**

Daily cycles were obtained for all the available wind records at all the stations (top panels in Fig. 5).
Consistent with previous analyses (Garreaud et al., 2011; Rahn et al., 2011), there was a clear diurnal
cycle in the wind speed, with higher intensities observed during the afternoon at the IP and PLV stations
and at night in TCR. During the night, the intensity of the wind decreases gradually until reaching a
minimum level in the morning (IP, PLV). At the Coquimbo station, the diurnal wind amplitude was
lower than at the other locations, but the wind intensity was greater in the afternoon and decreases at
night.

Similarly, the daily cycle of surface currents was obtained from all records at all the stations (top
panels, Fig. 6). In the exposed system (IP and TCR), a clear daily cycle of surface currents was observed.
The IP station stands out by having a daily cycle with a higher amplitude and intensity of currents during
the night, which were mainly directed toward the northwest. In the morning, the current begins to
weaken and gradually changes its direction toward the south, until the afternoon when the current is
directed toward the southeast. At TCR, the surface currents predominantly flow northward and were
more intense in the afternoon and at night. At the entrance of the bays (COQ and TON), a clear daily
cycle of surface currents was not observed. Instead, the currents were characterized by a predominant
southerly component that did not allow the diurnal variation of the surface currents to be identified, as it
was accomplished in the exposed system.

**3.5 Mean daily cycles of currents and their vertical structures**

From now on, we will focus on currents in the diurnal band (0.73 and 1.60 cpd) from the surface to
100 m. To compare diurnal currents during spring and winter periods, we obtained the vertical structure
of the diurnal currents variance (for $u$ and $v$ components) for each location during both seasons (Fig. 8).
In this analysis, November and July were selected as the representative months of spring and winter
(only July 2010 was common in all sites). The IP station had the highest variance in the surface layer for
both diurnal current components, which were far more intense in spring (November, 2009) than in



winter (July, 2010). At TCR, the variance of the meridional diurnal current component was larger than
those of the zonal component, and there were larger in spring (November, 2009) than winter (July, 2010).
At the entrance of the bays, there were no major differences in the variance of the meridional
components of spring (November, 2010) and winter (July, 2010). Unlike the sites exposed to the wind,
the variance of the zonal component of the diurnal currents in the surface layer of Coquimbo Bay was
slightly higher. This increase agrees with the slight increase in the zonal component of the wind that is
due to the sea breeze (Fig. 1c) (Garreaud et al., 2011; Rahn et al., 2011).

We have shown that the diurnal currents recorded at IP and TCR had the highest variance; for this
reason, we will focus on analyzing in more detail the stations of the exposed system. Figure 9 shows the
daily cycle of wind and currents during a period of spring (November) and winter (July) as a
representative of periods of higher and lower intensities in the seasonal wind cycle, respectively. For
each period, we estimated a referential depth of the wind effect on the sea surface friction layer which
was calculated following Eq. (1) (Stewart, 2007):

$$D_E = \frac{7.6}{\sqrt{sin|\varphi|}} \, U_{10} \,, \tag{1}$$

where $D_E$ is the depth of the Ekman layer,    is the latitude, and $U_{10}$ is  the wind speed. We
estimated $D_E$ with the mean wind speed for each period previously defined (Table 4). The IP station was
located in the most oceanic region where wind forcing is more intense. During spring the $D_E$ was ~54 m,
in winter the mean wind speed is lower at both sites (IP and TCR), and estimated $D_E$ was ~ 30 m.

In the case of intense winds (November), the IP station recorded the highest diurnal current
amplitude. At this location, the maximum of the meridional wind occurred at 21 h (local time) and the
maximum of the meridional surface current (8 m) responded with a delay of ~4 h (Fig. 9a, b). The
vertical structure of the daily cycle of the diurnal currents was characterized by an intensification of the
currents toward the north (south) during the night (day) that was associated with a component away
from (toward) the coast that extends down to approximately 50 m (Fig. 9b, c). In July 2010, a surface
layer 40 m deep was observed that followed this pattern but with lower intensities.

At TCR, for the period of intense winds (November, 2009), the maximum of the meridional wind



occurred at 24 h (local time), and the maximum of the meridional surface current (12 m) did not present
a phase lag. The vertical structure of the daily cycle of currents for the first 40 m depth was similar to IP
but with lower amplitudes. This daily cycle can be represented by the simple two-layer model forced by
the diurnal wind in which the surface layer extends up to 40 m and the bottom layer is phase-shifted by
180° as described above and as used by Sobarzo et al. (2010) in their study of the Concepcion Bay (37°S)
during the spring of 2007. Likewise, the study by Simpson et al. (2002) described that the vertical
structure of the current on the Namibian shelf (28.6°S) can also be described by this two-layer system,
where the surface layer flows in a direction opposite to that of the bottom layer.

**3.6  Synoptic variability of wind and its impact on diurnal currents**

Next, we will analyze whether the synoptic variability of wind influences diurnal currents; for this
reason, the wind series were separated into diurnal and synoptic bands. The synoptic wind variability
modulates its diurnal amplitude, i.e., an increase in (or a relaxation of) the wind manifests as an increase
(or decrease) in the amplitude of the daily cycle (Fig. 10a and 11a). This relationship was also observed
by Sobarzo et al. (2010) for different time periods in a wind series from Concepcion (37°S), California
(Rosenfeld, 1988), and the west coast of Australia (Pattiaratchi et al., 1997). The synoptic variations of
the wind also control the day-to-day variability of the surface currents, as documented by Garreaud et al.
(2011) in the Coquimbo Bay region. They identified intense southerly (northerly) winds favorable to
upwelling (downwelling) that generated a strong equatorward (southward) flow.

During periods of intense wind that last between 5 and 15 days, increases in the diurnal current
amplitude were observed at IP (Figs. 10 c-d), but when the wind suddenly weakens, in some cases these
amplitudes are maintained for 2 to 3 days. For instance, as occurred between 5 and 8 September 2010,
the increase in amplitude of the diurnal currents indicates the presence of inertial oscillations in response
to the decrease in the wind stress. In our study region, the diurnal and inertial frequencies are very
similar to each other; therefore, in those periods in which amplitude of the diurnal current increases
when wind stress decreases, the observed current energy would correspond to the diurnal-inertial
oscillations. Spectral estimates of surface buoy trajectories indicated a higher diurnal or near-inertial
energy during summer than winter (results not shown), confirming the results found for the surface
currents by moorings deployed at the exposed sites. These surface currents increased in the diurnal band,
which were frequently maintained for approximately 2 to 3 days, resulting as the response from a sudden





decrease in wind stress (Pollard, 1970). Pollard and Millard (1970) reported that wind favorable to the
generation of inertial oscillations would result from sudden changes in wind intensity and/or direction.
This pattern had been described by Shearman (2005) on the New England shelf (~41°N), who identified
that high current variability in the near-inertial band tends to be quite strong in summer and weak in
winter.

It has been shown that there was high diurnal variation in the currents of the exposed zone and

that coupling of these currents may occur due to the effects of diurnal wind forcing and inertial currents
on the continental shelf, as the latitude of the study area is close to an inertial frequency of $\sim 2*pi/24$ hr$^{-1}$
(Hyder et al., 2002; Simpson et al., 2002).  However, some of the diurnal band variability may be
explained by internal gravity waves (IGWs) due to the variation of the inertial frequency (or inertial
period) due to horizontal current gradients (Kunze, 1985; Shearman, 2005; Lerczak et al., 2001). In
general, current shear may affect the inertial frequency ($f$) and change the dynamics of IGWs (Kunze,
1985; Shearman, 2005); that is, spatial gradients of the currents may change vorticity, causing the
effective inertial frequency to approach or move away from the diurnal frequency. This may affect the
diurnal variation in our study area, considering that low-frequency current variability (from 5 to 90 days,
especially at 50 days) is dominated by coastal-trapped waves (CTW) of remote origin (Shaffer et al.,
1997; Hormazabal et al., 2002) that when propagated through the region, develop a zonal gradient in the
meridional velocity. The IGW theory states that in a continuously stratified ocean, IGWs may exist if the
following condition is met: $f < w < N$, where $f$ and $N$ are the inertial frequency and buoyancy,
respectively (Garrett and Kunze, 2007). Additionally, some studies of upwelling systems indicate that
intense internal wave events occur during periods of wind relaxation, i.e., when coastal upwelling ceases
and stratification intensifies (Lerczak et al., 2003; Aguirre et al., 2010; Bravo et al., 2013), which could
also affect the propagation of IGWs and the variability of the diurnal currents.

To assess to what extent coastal-trapped waves affected the inertial frequency, "f_effective" ($f_{eff}$)

was estimated using the results at 30.31°S of Shaffer et al. (1997), who estimated the zonal structure of
the meridional velocity of the first baroclinic mode of coastal-trapped waves (according to Brink, 1982).
The effective Coriolis frequency is obtained from $f_{eff} = f + (\partial v/\partial x)/2$ (Kunze, 1985; Nam and Send,
2013). When considering a negative (positive) phase in sea level, that is, negative (positive) anomalies
near the coast, the alongshore velocity structure of the first baroclinic mode CTW is positive (negative)
or northward (southward) on contiental shelf, then at 30°S the $f_{eff}$ changes 2% (-2%) with respect to the





Coriolis frequency or inertial period. At this latitude, the inertial period is 24 h, with the propagation of
CTW the effective inertial period would be 24.5 h in negative phase or 23.5 h in positive phase. This
indicates that inertial oscillations are generated to the south (but not so far) of the study area, which have
a shorter inertial period but close to 24 h, these oscillations can propagate in the area as internal gravity
waves, disturbing the diurnal signal of the currents and its vertical structure, because they propagate
northward diagonally.

In TCR, currents in the diurnal band has a more complex behavior with respect to the diurnal
forcing of the wind and its synoptic modulation. Only some periods can be associated with inertial
oscillations. Note that the lower variance of the diurnal currents in the u-component is due to the
topographical restriction due to the proximity of the coast (~ 2 km). As in IP, part of the variability of
the diurnal currents in TCR can be attributed to the change of the inertial period by the shear of the
CTW low frequency currents. It can also affect the change in the stratification due to the fact that TCR is
an important upwelling center in the region. Although IGWs may affect the diurnal variability in the
region when there is intense current shear (Kunze, 1985; Shearman, 2005), these are not analyzed in this
study; nevertheless this encourages us to continue studying the variability of diurnal currents in the area
in the near future.

**3.7  Case study: wind-driven diurnal currents in spring**

In the above sections, we have emphasized the high variability of diurnal currents in the study area,
reaching amplitudes of 30 cm s$^{-1}$ during the spring. In this section, we will focus on a specific time
period at Islote Pájaros that has been defined as a case study where the topographic and orographic
effects on wind and currents are less substantial (Bravo et al., 2016). Additionally, the mean amplitude
and variance of the diurnal current at this location were higher than those at the other stations. Wind data
at Islote Pájaros and data on currents at 8 and 72 m depths for the period between August 25 and
October 22, 2010 were selected. Complex demodulation was applied to the selected data (Pollard and
Miller, 1970; Simpson et al., 2002), and a 24 h harmonic fit was applied to the time series, with a
window of 48 h and a 4 h shift.

Figure 12 shows the amplitude and phase of the counterclockwise component of the wind (Fig. 12a,
b) and the currents at 8 and 72 m depths (Fig. 12c, d). During the spring of 2010, the diurnal wind was





quite strong, with a high temporal variability and a marked counterclockwise rotation (CCW). Similarly,
the diurnal current had a predominantly CCW rotation that reached amplitudes of 25 cm s$^{-1}$.
Additionally, there is a strong relationship between the surface current stress and the wind, mainly
between the 13$^{th}$ and 18$^{th}$ of October, where it was clearly seen that the current was forced by the diurnal
wind. This analysis for the case of Talcaruca (not shown) was less clear, given that the system is more
complex due to several factors, changes in stratification, shear of low frequency currents by CTW and
the presence of internal gravity waves.

Lerczak et al. (2001), Hyder et al. (2002), Simpson et al. (2002), and Hyder et al. (2011) indicated
that much of the increase in the current amplitude is due to diurnal wind forcing. In IP, increases in the
intensity of the surface current were common during spring and summer. Wind data indicated that the
highest variability occurs during this period, and as we have noted previously, the amplitude of the daily
wind cycle is modulated by synoptic-scale circulation, i.e., diurnal wind forcing increases when the wind
favors coastal upwelling and decreases when the wind relaxes; this occurs mainly in spring and summer
(Rahn and Garreaud, 2013). Nevertheless, diurnal-inertial oscillations were observed during September
4$^{th}$ –7$^{th}$, September 23$^{rd}$–25$^{th}$, and October 2$^{nd}$–5$^{th}$. When the wind amplitude began to decrease, the
surface current began to increase and reached levels of 25 cm s$^{-1}$. During these events, the surface
current phase plot (Fig. 12d) showed a smooth variation of 5°/day (gray line), which was consistent with
the inertial period at the location (24.02 h). Simpson et al. (2002) indicated that this phase change is
consistent with an alternation between forced oscillation periods, where diurnal movements predominate,
and wind forcing decreases the generation of inertial oscillations.

At ~30°N/S, there may be resonance between the diurnal and inertial frequencies (Simpson et al.,
2002; Hyder et al., 2002, 2011), as diurnal wind forcing would generate an anticyclonic rotation that
would contribute to the energy at this frequency. If we compare the study conducted by Sobarzo et al.
(2010) with our results, we can see that inertial oscillations at 36.5°S have smaller amplitudes (~half)
than those observed near 30°S when the wind forcing in both regions that preceded the inertial
oscillations have similar magnitudes. At these latitudes, IGWs may also appear near the inertial band
(Lerczak et al., 2001; Alford et al., 2016) and propagate obliquely (from the bottom to the surface),
unlike inertial oscillations, which propagate horizontally. The vertical propagation of IGWs near the
inertial frequency favors the mixing process, which may be particularly important in the supply of
nutrients in coastal ecosystems (Lucas et al., 2014). These characteristics generate new questions for



future studies, such as whether there are spatial variations in the diurnal wind forcing that may influence
the spatial structure of the currents, especially in the propagation of waves near the inertial frequency.

**4   Summary**

In northern-central Chile (~30°S), current measurements were available at four sites with more than
one year of data. The area includes a coastal upwelling center exposed to the sea wind (Islote Pájaros
and Talcaruca) and a large system of bays (~100 km long) located north of the upwelling center
(specifically, the Coquimbo and Tongoy Bays). The most relevant findings are summarized below:

1 ) The circulation in the system exposed to the prevailing wind (TCR and IP) exhibits typical

characteristics of an eastern boundary current system over the continental shelf, with a surface

layer that is dominated by the wind and below a poleward subsurface current, consistent with

various studies previously conducted in Chile (Bakun and Nelson, 1991, Strub et al., 1998,

Shaffer et al., 1999, Aguirre et al., 2012). In the entrace to the Coquimbo and Tongoy bays, the

flows are consistent with recirculation patterns observed in previous studies (Valle-Levinson and

Moraga, 2006; Moraga et al., 2011).

2 ) The seasonal cycle of daily-mean surface currents at the surface is complex and not as marked as

that of the wind. In the wind-exposed zone, the intensity of surface current is higher in spring and

lower in winter, and its temporal variation shows signs of intraseasonal influence. At the

entrance to the bays, the surface currents do not exhibit a clear seasonal cycle but instead are

characterized by local recirculation patterns.

3 ) The surface currents in the exposed system showed significant energy in the diurnal band. This

indicates that the currents at this frequency were influenced by diurnal wind forcing and inertial

oscillations (Simpson et al., 2002; Hyder et al., 2011). In the exposed zone, the amplitude of

diurnal currents is stronger in spring and summer compared to autumn and winter. This was

consistent with Lagrangian measurements of the surface currents, which showed a higher diurnal

energy during summer than winter. The most intense currents developed at night and have a

predominant northward orientation that begin to weaken during the morning. At the entrance to



the bays, the amplitude is lower, with a predominant meridional component throughout the day,
indicating that surface currents are dominated mainly by the local recirculation patterns in each
bay.

4) The vertical structure of the amplitude of the diurnal currents in the exposed system is
represented by a surface layer which is mainly influenced by the wind. The surface layer is
characterized by the intensification of currents toward the north (south), associated with a
component away from (toward) the coast during the night (day) that responds to daily wind
variations due to sea breeze. The surface layer is deeper in spring and has diurnal currents with
larger amplitudes as the response of stronger sea breeze during the season.

5) Diurnal wind variability is modulated by synoptic-scale circulation (3 to 15 days), which directly
affects the diurnal current response. Under upwelling-favorable wind conditions, diurnal wind
forcing occurred mainly by daily wind variations due to the sea breeze, while a sudden decrease
in wind intensity generated inertial oscillations. Because the study area is located near the critical
latitude of 30°S, inertial motions have a period of ~24 h, which is similar to those of diurnal
wind and tidal forcings (Simpson et al., 2002; Hyder et al., 2011). However, the diurnal tidal
component has a small amplitude in the region (Bravo et al., 2013). The highest diurnal current
variability occurs at Islote Pájaros, located ~22 km from the coast, indicating that a coupling may
exist between diurnal wind forcing, inertial oscillations, and IGWs near the diurnal band (Alford
et al., 2016) that may affect the diurnal variation in the region when there is intense current shear
(Kunze, 1985; Shearman, 2005).


**Competing interests**
The authors declare that they have no conflict of interest.

**Acknowledgements**
This work was partially financed by CONICYT grant no. 21090292 and Center for Climate and
Resilience Research (CR2, CONICYT/FONDAP/15110009). M. Ramos acknowledge support from
FONDECYT (projects: 1080606 and 1140845) and from INNOVA-CORFO (07CN13 IXM-150). M.



Ramos, L. Bravo and M. Thiel acknowledge support from Chilean Millennium Initiative (NC120030).
M. Thiel acknowledge support from FONDECYT (project: 1100749). L. Bravo acknowledge support
from CONICYT-PAI ("Concurso Nacional Inserción en la Academia, Convocatoria 2016, Folio
79160044").

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



**Figure captions**

**Figure 1.** Study area with bathymetry and localization of current deployments ADCP-RDI, mean
circulation and ellipses for currents (diurnal band). (a) Surface and (b) bottom layer. (c) Location of
meteorological and hodograph stations of daily surface wind cycle. Black crosses indicate the origin  ($v_x$
$= v_y = 0$). All hodographs have anticyclonic gyres.

**Figure 2.** Vertical profile of temporal averaged current (black line) and standard deviation (thin line). (a)
U component and (b) V component. (c) Polarization of diurnal current in the stations of Islote Pájaros
(IP), Talcaruca (TCR), Coquimbo Bay (COQ) and Tongoy Bay (TON). Average data periods
correspond to all data available at each station (see Table 1).

**Figure 3.** Rotational spectra of wind: (a) Islote de Pájaros (IP) and (b) Talcaruca (TCR). Rotational
spectra of total surface current (black line: counterclockwise, red line: clockwise) in the stations of (c) IP
(8m) and (d) TCR (12m).

**Figure 4.** Vertical variation of rotational spectral power of the current (upper panel (a):
counterclockwise, bottom panel (b): clockwise) at the Islote Pájaros (IP), Talcaruca (TCR), Coquimbo
Bay (COQ) and Tongoy Bay (TON).

**Figure 5.** (a) Mean daily cycle of wind magnitude (red line), (b) Temporal variation of daily wind cycle
and monthly mean wind magnitudes (right panel: blue line) at Islote Pájaros (IP), Talcaruca (TCR),
Coquimbo (COQ) and Punta Lengua de Vaca stations (PLV) from 2009 to 2010.

**Figure 6.** (a) Mean daily cycle of magnitude of the total surface current (red line), (b) Temporal
variation of the daily cycle of total surface current and monthly mean of total surface current magnitude
(right panel: blue line) at Islote Pájaros (IP), Talcaruca (TCR), Coquimbo Bay (COQ) and Tongoy Bay
(TON) from 2009 to 2010.

**Figure 7.** Buoy trajectories in the Coquimbo bay area. (a) Summer (January, 2012), (b) winter (July,

2011).






**Figure 8.** Variance of diurnal currents, (a) *v* component, (b) *u* component, for all periods (black line),
intense (red line) and weak (blue line) period of wind in the stations of Islote Pájaros (IP) and Talcaruca
(TCR).

**Figure 9.** Daily cycles, (a) daily wind (thick line: *v* component, thin line: *u* component), (b) *v* component
and (c) *u* component of diurnal currents, intense and weak period of wind in the stations of Islote Pájaros
(IP) and Talcaruca (TCR). Months used to calculate the daily cycle of currents during a period of intense
wind in November 2009 and for a less intense period that was in July 2010.

**Figure 10.** Islote Pájaros station (IP), (a) meridional wind stress, (b) zonal wind stress, (c) *v* component
and (d) *u* component of the diurnal currents. Segmented vertical lines indicate increased amplitudes of
diurnal currents when the wind increases and/or it remains intense.

**Figure 11.** Talcaruca station (TCR), (a) meridional wind stress, (b) zonal wind stress, (c) *v* component
and (d) *u* component of the diurnal currents. Segmented vertical lines indicate increased amplitudes of
diurnal currents when the wind increases and/or it remains intense.

**Figure 12.** Amplitude and phase of the CCW component of wind (a-b) and current (c-d) at 8m (black
line) and 72m (gray line) depth obtained in IP from complex demodulation. The solid gray line
represents the phase change (5°/day) for pure inertial movement.





**Table titles**

**Table 1:** Oceanographic and meteorological information available in this study.

**Table 2:** Results of the fit to the wind time-series using the least square method of a 24-h harmonic. U:
zonal component, V: meridional component. IP: Islote Pájaros, TCR: Talcaruca, COQ: Coquimbo, PLV:
Punta Lengua de Vaca.

**Table 3:** Results of the fit to the surface currents time-series using the least square method of a 24-h
harmonic. U: zonal component, V: meridional component. IP: Islote Pájaros, TCR: Talcaruca, COQ:
Coquimbo Bay, TON: Tongoy Bay.

**Table 4**: Ekman layer depth ($D_E$).




























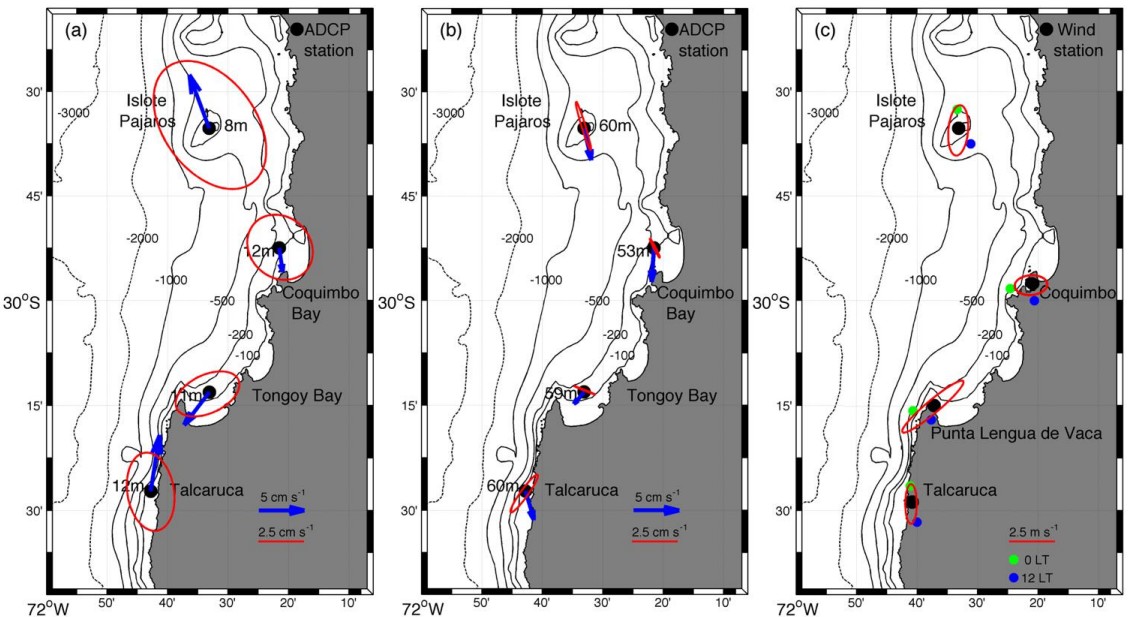


**Figure 1**. Study area with bathymetry and localization of currents stations ADCP-RDI, mean circulation
and ellipses currents (diurnal band). (a) Surface and (b) bottom layer. (c) Location of meteorological and
hodograph stations of daily surface wind cycle. Black crosses indicate the origin ($v_x = v_y = 0$). All
hodographs have anticyclonic gyres.







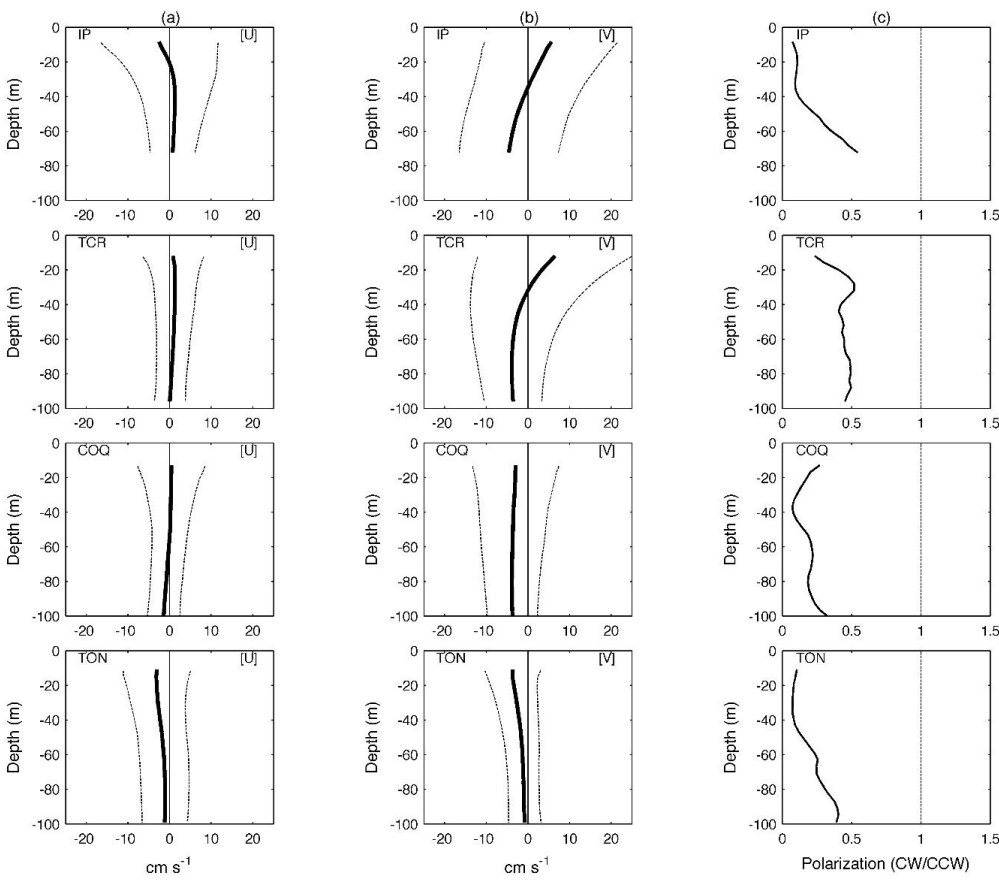


**Figure 2.** Vertical profile of temporal averaged current (black line) and standard deviation (thin line). (a) U component and (b) V component. (c) Polarization of diurnal current in the stations of Islote Pájaros (IP), Talcaruca (TCR), Coquimbo Bay (COQ) and Tongoy Bay (TON). Average data periods correspond to all data available at each station (see Table 1).




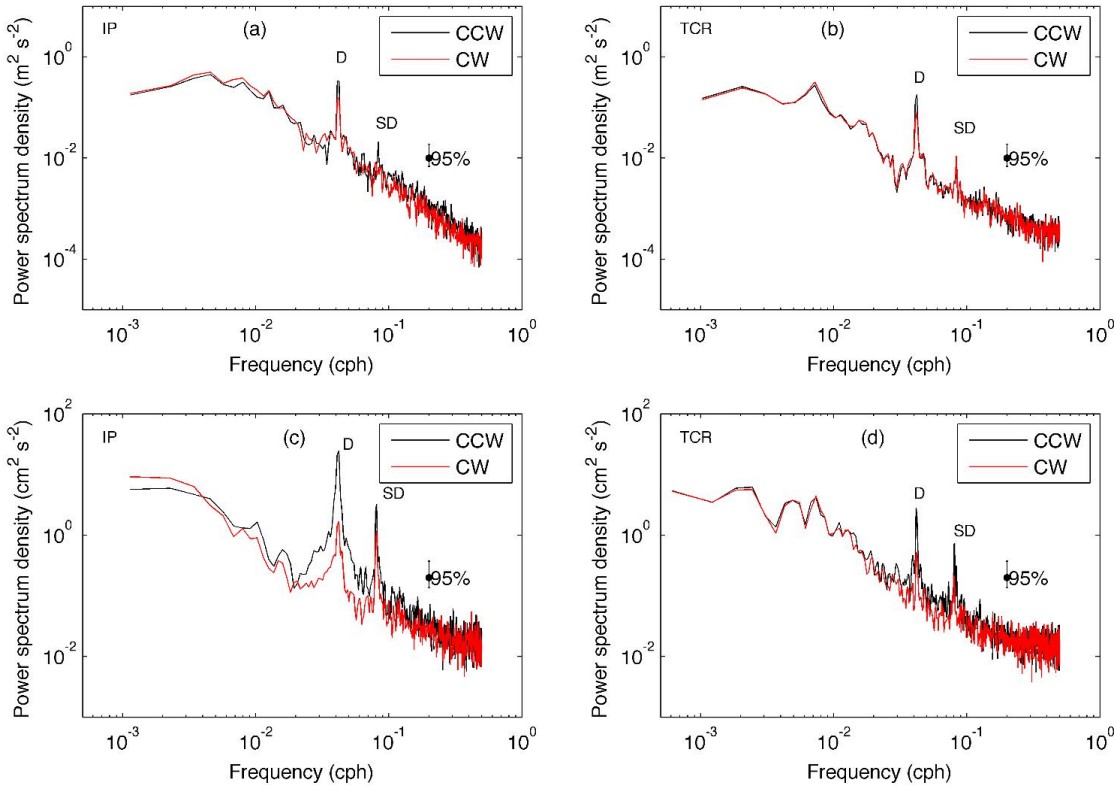

**Figure 3.** Wind rotational spectra: (a) Islote de Pájaros (IP) and (b) Talcaruca (TCR). Rotational spectra
of total surface current (black line: counterclockwise, red line: clockwise) in the stations of (c) IP (8m)
and (d) TCR (12m).





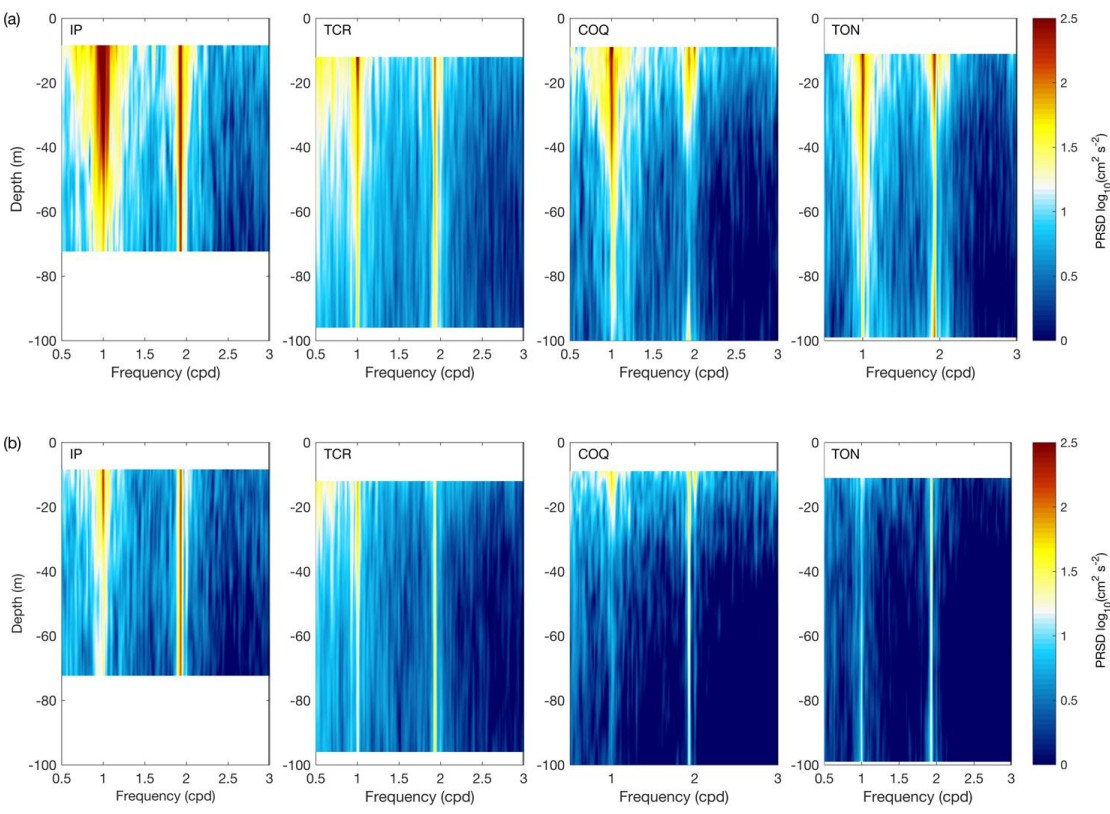


**Figure 4.** Vertical variation of rotational spectral power of the current (upper panel (a): counterclockwise, bottom panel (b): clockwise) at the Islote Pájaros (IP), Talcaruca (TCR), Coquimbo Bay (COQ) and Tongoy Bay (TON).




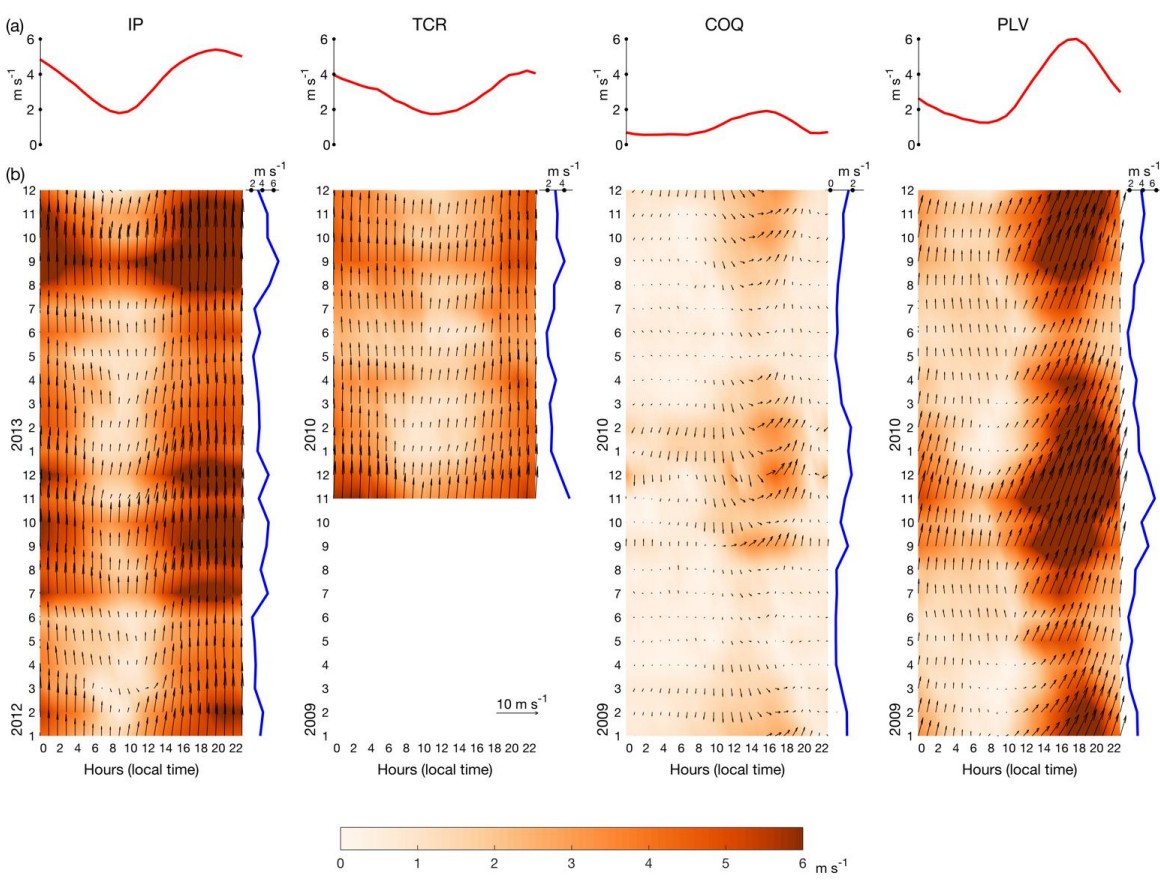

**Figure 5.** (a) Mean daily cycle of wind magnitude (red line), (b) Temporal variation of daily wind cycle
and monthly mean wind magnitudes (right panel: blue line) at Islote Pájaros (IP), Talcaruca (TCR),
Coquimbo (COQ) and Punta Lengua de Vaca stations (PLV) from 2009 to 2010.



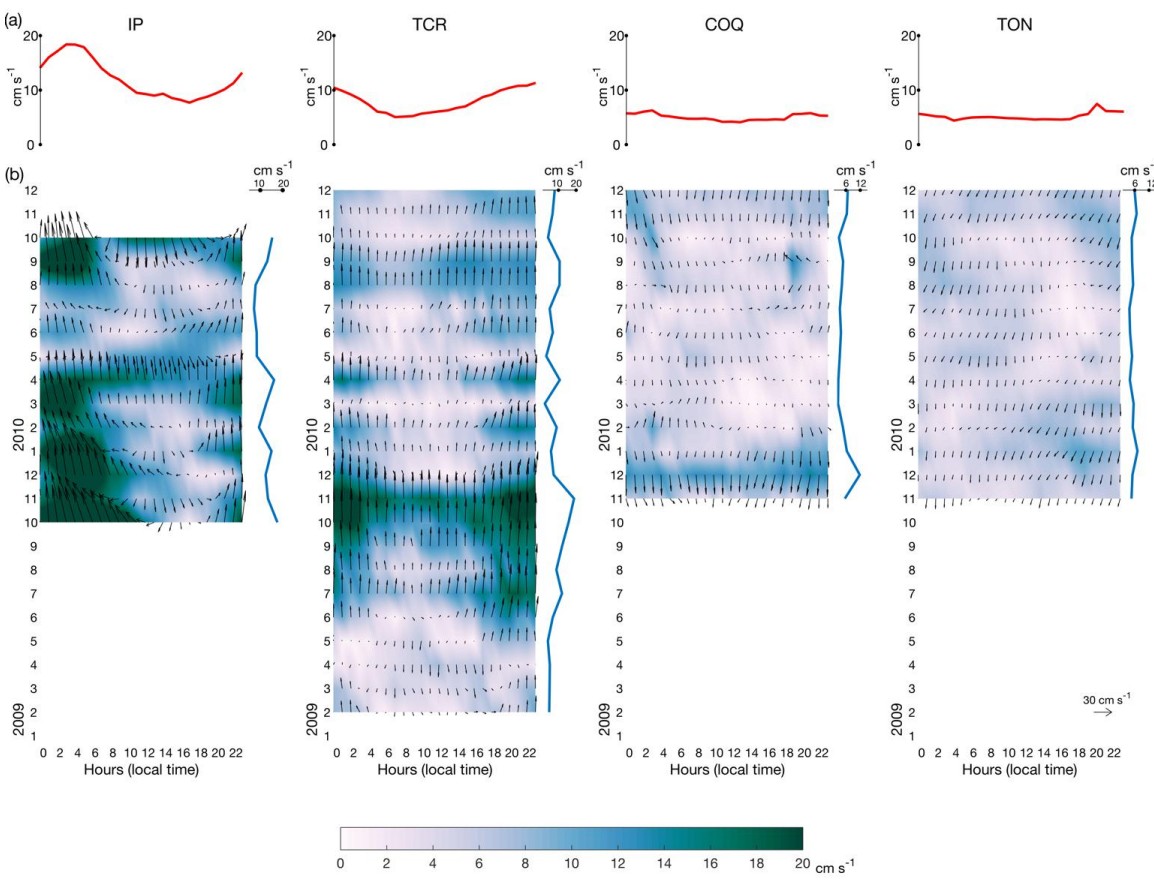

**Figure 6.** (a) Mean daily cycle of magnitude of the total surface current (red line), (b) Temporal
variation of the daily cycle of total surface current and monthly mean of total surface current magnitudes
(right panel: blue line) at Islote Pájaros (IP), Talcaruca (TCR), Coquimbo Bay (COQ) and Tongoy Bay
(TON) from 2009 to 2010.





(a)           (b)

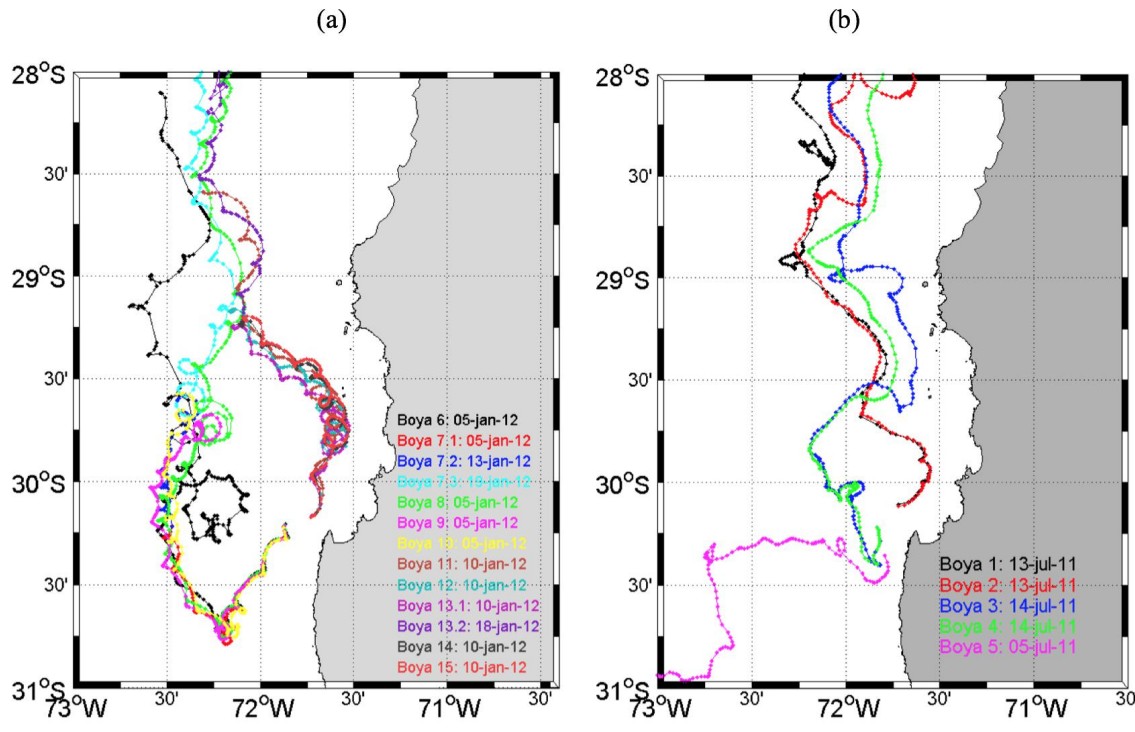


**Figure 7.** Buoy trajectories in the Coquimbo bay area. (a) Summer (January, 2012), (b) winter (July, 2011).




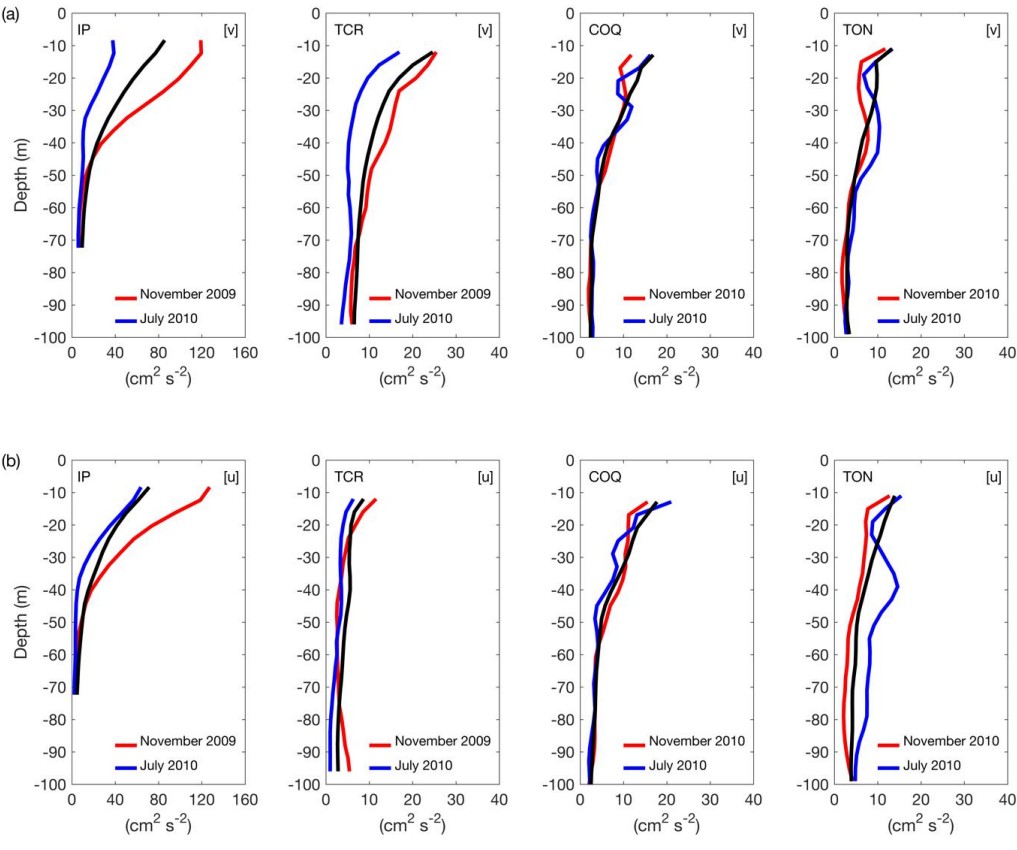

**Figure 8.** Variance of diurnal currents, (a) *v* component, (b) *u* component, for all periods (black line),
intense (red line) and weak (blue line) period of wind in the stations of Islote Pájaros (IP) and Talcaruca
(TCR).





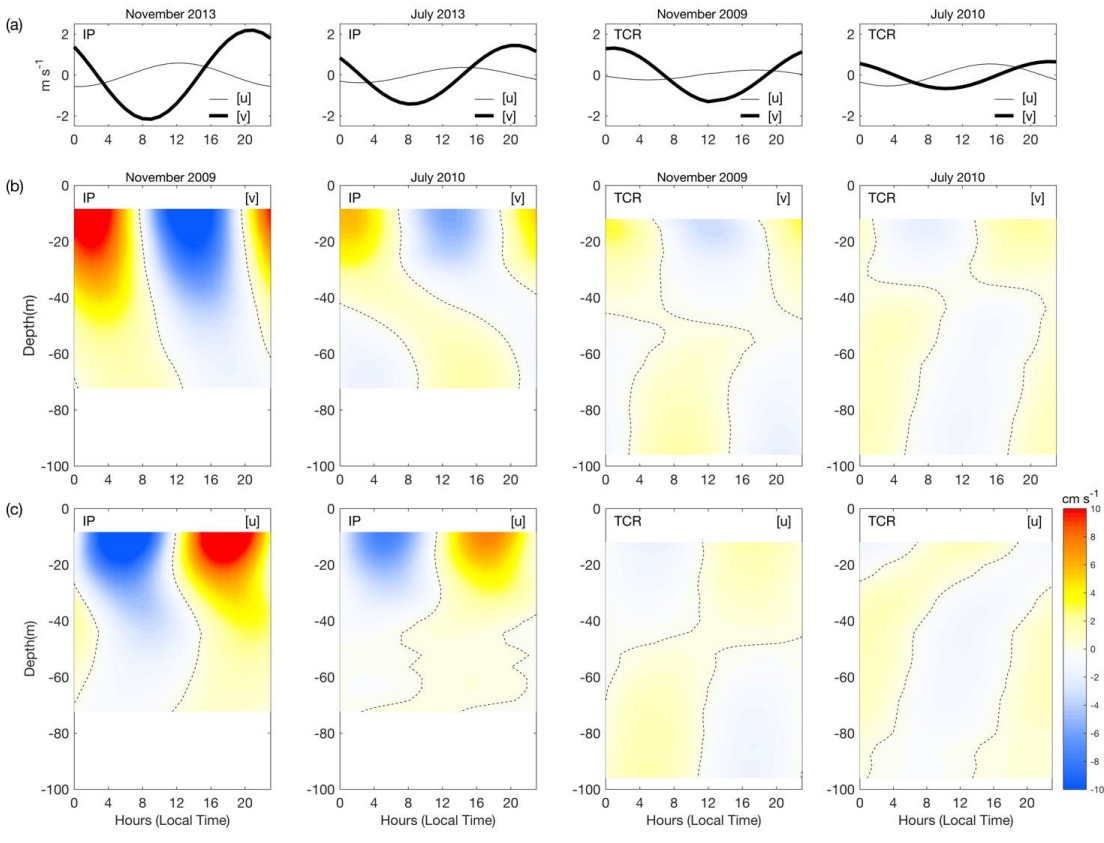


**Figure 9.** Daily cycles, (a) daily wind (thick line: *v* component, thin line: *u* component), (b) *v*

component and (c) *u* component of diurnal currents, intense and weak period of wind in the stations of

Islote Pájaros (IP) and Talcaruca (TCR). Months used to calculate the daily cycle of currents during a

period of intense wind in November 2009 and for a less intense period that was in July 2010.




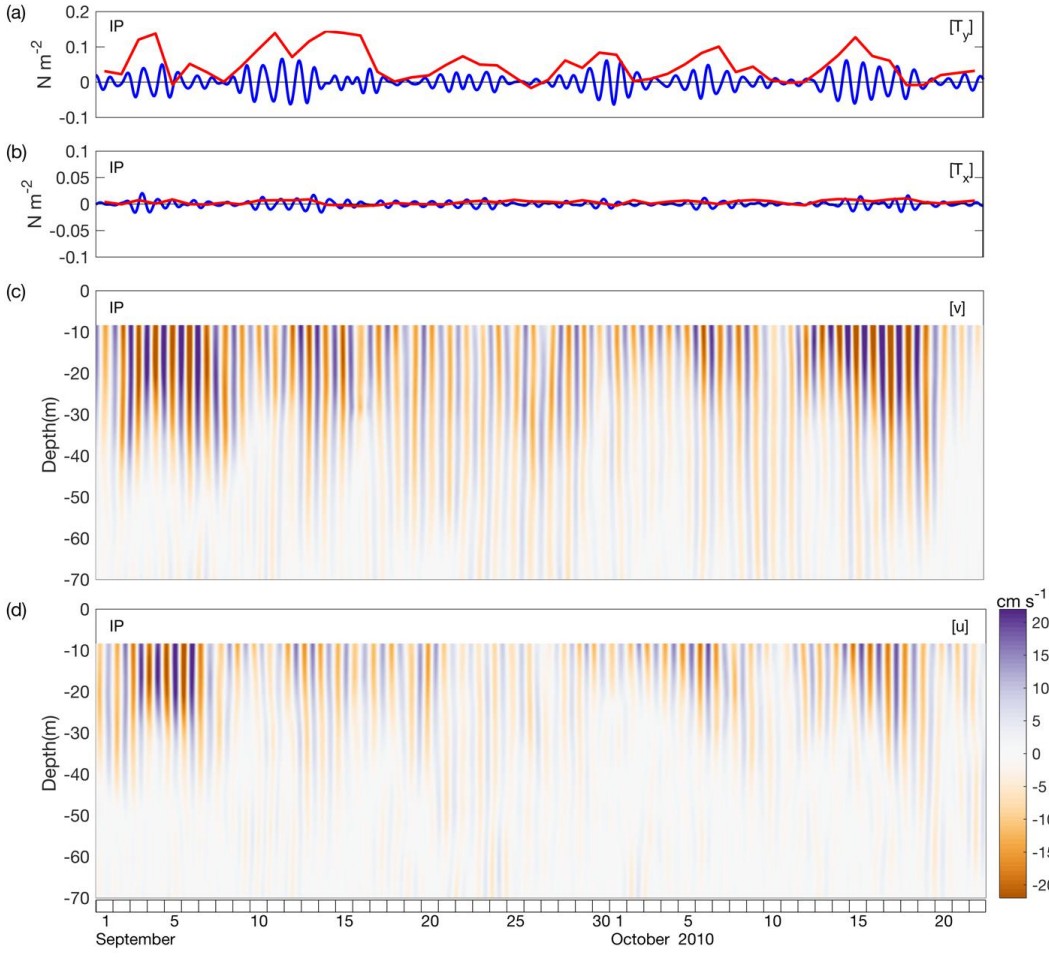

**Figure 10.** Islote Pájaros station (IP), (a) meridional wind stress, (b) zonal wind stress, (c) *v* component
and (d) *u* component of the diurnal currents. Segmented vertical lines indicate increased amplitudes of
diurnal currents when the wind increases and/or it remains intense.




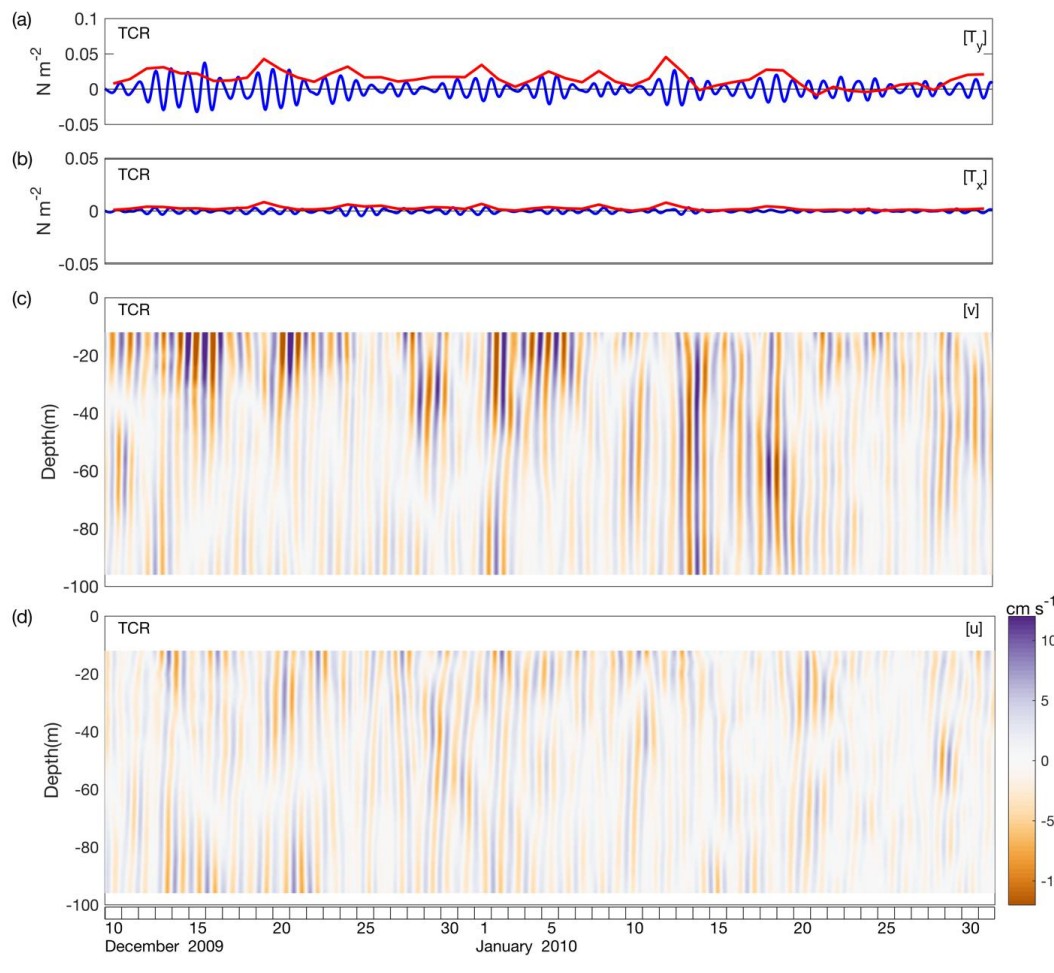

**Figure 11.** Talcaruca station (TCR), (a) meridional wind stress, (b) zonal wind stress, (c) $v$ component
and (d) $u$ component of the diurnal currents. Segmented vertical lines indicate increased amplitudes of
diurnal currents when the wind increases and/or it remains intense.




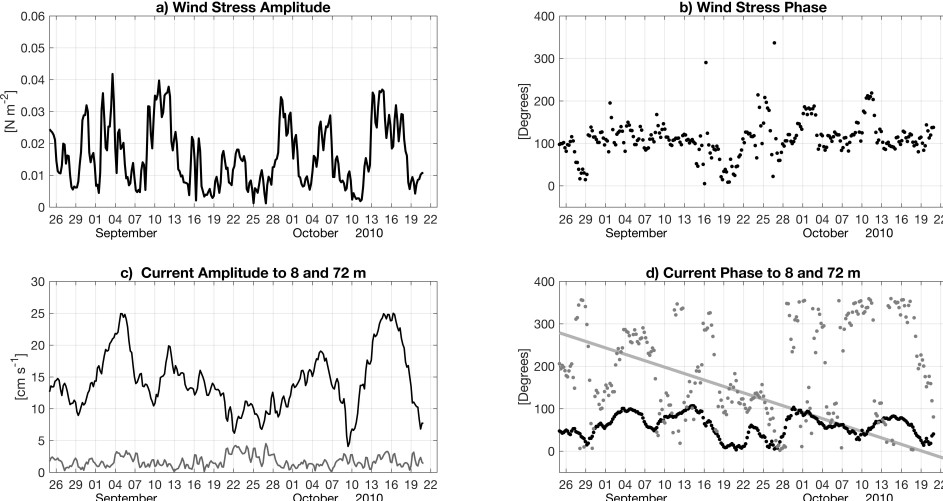

**Figure 12.** Amplitude and phase of the CCW component of wind (a-b) and current (c-d) at 8m (black
line) and 72m (gray line) depth obtained in IP from complex demodulation. The solid gray line
represents the phase change (5°/day) for pure inertial movement.


















**Table 1.** Oceanographic and meteorological information available in this study.

| a) Oceanographic stations /Instrument | Variables | Recording interval | Bottom depth (m) | Level (bin size) | Period |
|---|---|---|---|---|---|
| a. Islote Pájaros (29.58°S,71.55°W) | | | | | |
| ADCP | direction, speed | 30 min | 190 | 8.3-72.3 (4m) | 21 October 2009-23 October 2010 |
| b. Coquimbo Bay (29.87°S,71.36°W) | | | | | |
| ADCP | direction, speed | 30 min | 120 | 8.8-100.8 (4m) | 21 November 2009-30 December 2010 |
| c. Tongoy Bay (30.21°S,71.55°W) | | | | | |
| ADCP | direction, speed | 30 min | 120 | 11-99 (4m) | 21 November 2009-30 December 2010 |
| d. Talcarura (30.45°S,71.71°W ) | | | | | |
| ADCP | direction, speed | 60 min | 120 | 12-96 (4m) | 29 January 2009-11 December 2010 |

| b) Automatic weather stations | Variables | Period |
|---|---|---|
| Islote Pájaros (29.58°S, 71.55°W) | direction, speed | 05 August 2010-31 December 2010 |
| | | 01 January 2012-31 December 2013 |
| Coquimbo (29.96°S, 71.35°W) | direction, speed | 01 January 2009-31 December 2010 |
| Punta Lengua de Vaca (30.25°S, 71.62°W) | direction, speed | 01 January 2009-31 December 2010 |
| Talcaruca (30.48°S, 71.70°W) | direction, speed | 21 November 2009-31 December 2010 |






**Table 2.** Results of the fit to the wind time-series using the least square method of a 24-h harmonic. U:
zonal component, V: meridional component. IP: Islote Pájaros, TCR: Talcaruca, COQ: Coquimbo, PLV:
Punta Lengua de Vaca.

| | Amplitude (m s⁻¹) | | Phase (deg) | | % of explained variance | |
|---|---|---|---|---|---|---|
| Station | U | V | U | V | U | V |
| IP | 0.63 | 1.64 | 64 | 152 | 12.07 | 7.05 |
| TCR | 0.34 | 1.17 | 75 | -176 | 13.80 | 6.85 |
| COQ | 1.04 | 0.47 | 51 | 134 | 28.17 | 4.32 |
| PLV | 2.10 | 1.43 | 90 | 117 | 32.01 | 13.28 |





**Table 3.** Results of the fit to the surface currents time-series using the least square method of a 24-h
harmonic. U: zonal component, V: meridional component. IP: Islote Pájaros, TCR: Talcaruca, COQ:
Coquimbo Bay, TON: Tongoy Bay.

| | Amplitude (m s$^{-1}$) | | Phase (deg) | | % of explained variance | |
|---|---|---|---|---|---|---|
| Station | U | V | U | V | U | V |
| IP | 7.72 | 8.58 | -31 | 85 | 14.76 | 14.34 |
| TCR | 1.64 | 3.53 | 64 | 152 | 2.50 | 1.83 |
| COQ | 2.50 | 3.48 | 52 | 110 | 4.29 | 5.78 |
| TON | 1.64 | 2.25 | -62 | 36 | 2.02 | 5.43 |



**Table 4.** Ekman layer depth ($D_E$).

| Station | spring | | winter | |
|---------|-----------------|-----------|-----------------|-----------|
| | $U_{10}$ (m/s) | $D_E$ (m) | $U_{10}$ (m/s) | $D_E$ (m) |
| IP | 5 | 54 | 2.5 | 27 |
| TCR | 4.5 | 48 | 2.7 | 28 |