# Peer review of "Seasonal and synoptic variability of diurnal currents in an upwelling system off northern Chile near 30°S"

_Ocean Science, 2018_

## Referee Comment (RC1) · Anonymous Referee #1 · 23 May 2018

In their manuscript, Monica Bello and co-authors analysed the variability of diurnal currents off northern Chile.

The main problems I had with this work are presented first, followed by a list of other issues.

1. What are the new scientific findings? When reading the text, it seems as if almost every findings replicates or is very similar to previous research findings. You should clearly state in the abstract and in the conclusion the new findings of your study.

2. Source of data It is quite common to properly disclose the data sources. Who collected the data? How do I get access to the data? Given these are older data sets,

you should summarize all research publications that made use of this particular data set and highlight what is new in your application.

3. Tides While this is about diurnal currents, it is quite remarkable to note that diurnal tides are not mentioned in the text until the last point of the summary. Tides truly need to be discussed as part of the data analysis.

4. IGW and CTWs Your study exclusively uses current and wind data. Despite this, you discuss potential influences by internal gravity waves and coastally trapped waves in the absence of any direct observational evidence of density effects. These parts are way too speculative in my opinion and should be removed from the text, unless they can be directly linked to your data.

5. Your Summary Here you should highlight only the new findings of your scientific investigation (and add recommendations for future research). However, your summary actually refers to some points that you didn't consider in your study at all, such as tides. This includes your statement that "the surface layer is deeper in spring". While you estimated the thickness of the surface Ekman layer, you haven't measured the true depth of the surface mixed layer from CTD data, have you?

6. What creates inertial oscillations? In the 4th point of your summary, you state that inertial oscillations are generated by a sudden decrease in wind intensity. This statement is incorrect. You need a sudden increase (not decrease) in wind intensity. For instance, a decrease in wind intensity has little impact on the dynamics of ocean processes that are geostrophically adjusted.

Here are the other issues in chronological order:

1. In the abstract and elsewhere you state that "the diurnal wind variability is modulated by the synoptic scale circulation". You probably refer to the synoptic wind patterns? Please add the term "atmospheric" before "circulation"

2. Line 66/67: There is a "(Ramos, pers. comm.)". Is this the same Ramos, who is also

co-author? It would be quite unusual to have a personal communication by a co-author included in a paper that is actually co-written by the same person.

3. Line 72/73: "Diurnal wind forcing (. . .) can equalize the diurnal and inertial frequencies". I don't understand this sentence. Please rewrite.

4. Line 95. "report" => change to "paper"

5. Line 146: Can you describe the method of "complex demodulation" in more detail.

6. In section 3.2 you refer to semidiurnal currents and you make several statement including the words "could be resonating" and "may be explained". In my view these statement should be deleted or merged with similar statements made in the introduction.

7. Line 230: You compare the hodographs with "anticyclonic gyres". This terminology is grossly incorrect.

8. Line 272: You state that "the maximum current intensities agreed with the maxima of the wind". What do you mean? Average wind speed? Diurnal wind-speed maximum?

9. Line 281: Here you refer to satellite-tracked surface buoys without giving any references. Who did the deployments? More details on this should be given in the methodology section, not in the results section.

10. Line 284: What do you mean by "maximum winds"? Do you mean the strongest synoptic wind events? Or the strongest sea breezes?

11. Line 339: Estimate of the Ekman layer thickness (equation 1) should be included in the methodology section not here. Note that the true mixed layer depth can differ from the Ekman layer depth.

12. Line 447: Why a "4 h shift"? I am totally lost here.

---

## Referee Comment (RC2) · Anonymous Referee #2 · 5 Jun 2018

General comments:

This paper is an attempt to exhibit the seasonal and synoptic variabilities of near-inertial oscillations (NIOs) in the upwelling region of Chile. In-situ current data and wind measurements are used to establish the results. However, the paper is sort of reproducing or replicating the scientific findings in previous studies of this research field, and nearly every finding is consistent with the previous results in the same region. So far, the scientific content of this paper seems not new and not compelling, so the intellectual or scientific merits of this study need to be reconsidered, actually identified, and hence highlighted.

Second, this paper lacks physics as a good physical oceanography paper should have. The physics associated with this topic can be very compelling. The coastal ocean off Chile is an ideal place to investigate near-inertial waves (NIWs) in a strongly baroclinic regime because of its unique upwelling system. Also, coastally trapped waves (CTWs) plays an important role in this region (as the authors mentioned), so the interaction between low-frequency waves (CTWs) and high-frequency waves (NIWs) may modify the propagation of NIWs. Anyway, I believe this study can be conducted deeper and further by adding more physics rather than simply reexhibiting phenomena of a well-known physical mechanism. That way this paper can be published in a high impact journal like Ocean Science, and, most importantly, push forward the frontier in this research field.

Last, this paper has problems on establishing comprehensive and systematic results. The record length of the current measurements is too short (about 1 year) to generalize the seasonal variability of NIOs. Usually, some sort of data with a length of multiple years or even a decade is necessary to establish a seasonal variability result. So, the seasonal variability finding is only specific for that year, but cannot be generalized. In addition, for establishing the synoptic variability, the paper doesn't have a figure and a comprehensive description to show/explain the mechanism/process about how CTWs (stimulated by the synoptic wind) modify the NIWs/NIOs and hence affect the synoptic variability of NIOs/NIWs. In other words, this paper lacks physical significance in addressing the synoptic variability issue.

Specific comments:

Line 128 – 130: Why is the complex spectral analysis only performed to IP and TCR? Why not include COQ and PLV?

Line 140 – 141: Tide might be an issue in this study. Is it possible to remove the tidal signals from the current measurements? That way the authors can safely establish results.

Line 198 – 200: According to Fig. 3, the counterclockwise and clockwise spectra seems on the similar order at the diurnal band. Why "the diurnal band had predominantly counterclockwise rotation"?

Line 204 – 206: Why use the data at 8m (IP) and 12m (TCR)? I guess that might be associated with the quality control. Please demonstrate this explicitly in the data subsection. Actually, the data subsection needs lots of improvements, since this study is heavily based on observations.

Line 245 – 247: Define the polarization.

Line 258 – 260: I found that the studies by Garreaud et al., 2011 (cited by 8 times) and Rahn et al., 2011 (cited by 7 times) are hugely influential for this study. If they are really important, summarize them in the introduction. Highlight what the new findings are in the Results Section.

Line 265 – 269: CTWs are important agents in this region. How did the CTWs get stimulated? What are the features of the CTWs propagating along the coast of Chile? Also, showing the fluctuations of sea level or other properties caused by CTWs is very important to help readers better understand how the CTWs influence the observed flow direction change in May 2010.

Line 368 – 369: I am confused about the "the amplitude of the daily cycle (Fig. 10a and 11a)". Is Fig. 10a and 11a showing the original data or the decomposed data at the diurnal band? If they are the decomposed data, are they anticyclonic or cyclonic?

Line 296 – 398: IGWs are too general for this study. Near-inertial waves or near-inertial internal gravity waves would a better and more accurate terminology for this study compared to the IGWs, since this paper is focusing on the diurnal cycle which is near-inertial at 30S. Review more studies about NIWs to fit the topic rather than reviewing the classic IGWs.

Line 412 – 424: Upwelling is an important feature in this region, which induces lateral

density gradients. Not only the shearing vorticity but also the lateral density gradients can modify the properties of NIWs. It can make this study unique. Take a look at the 2013 JPO paper by Whitt and Thomas.

Whitt, D. B., and L. N. Thomas, 2013: Near-Inertial Waves in Strongly Baroclinic Currents. J. Phys. Oceanogr., 43, 706–725, doi:10.1175/JPO-D-12-0132.1.

Line 453 – 454: What is "surface current stress"?

Line 473 – 475: More physics associated with the resonantly-forced inertial motions is definitely needed here or in other related subsections to make it be a good physical oceanography paper. For the fundamentals to start with, I recommend two recent papers about the resonantly-forced inertial motions and associated energy transfer across scales.

Whitt, D. B., and L. N. Thomas, 2015: Resonant Generation and Energetics of Wind-Forced Near-Inertial Motions in a Geostrophic Flow. J. Phys. Oceanogr., 45, 181–208, doi:10.1175/JPO-D-14-0168.1.

Thomas, L. N., 2017: On the modifications of near-inertial waves at fronts: implications for energy transfer across scales. Ocean Dyn., 67, 1335–1350, doi:10.1007/s10236-017-1088-6.

Fig. 1 caption: Unclear caption. What are the green and blue dots? What does "Black crosses indicate the origin (vx = vy = 0)" mean? Where are the "Black crosses"? What are "0 LT" and "12 LT"?

Fig. 10 caption: What do the blue and red lines represent? Line legend is missing. Same problem in the caption of Fig. 11.

Technical corrections:

Fig. 2 caption: Grammar issue. "temporal averaged" -> "temporally averaged".

Fig. 3 caption: Grammar issue. "in the stations" -> "at the stations". Same problem in

the captions of Fig. 8 and Fig.9.

Fig. 4 caption: Grammar issue. Use plural. Same problem in the caption of Fig. 9.

Fig. 8 caption: Grammar issue. "intense (red line) and weak (blue line) period of wind" -> "intense (red lines) and weak (blue lines) periods of wind".